analytical chemistry

*Dendrobium*, mid-infrared spectroscopy, near-infrared spectroscopy, chemometrics, authentication

**Authors for correspondence:**
Heng-Yu Huang
e-mail: hhyhhy96@163.com
Yuan-Zhong Wang
e-mail: boletus@126.com

This article has been edited by the Royal Society of Chemistry, including the commissioning, peer review process and editorial aspects up to the point of acceptance.

# Original plant traceability of *Dendrobium* species using multi-spectroscopy fusion and mathematical models

Ye Wang[1], Zhi-Tian Zuo[2], Heng-Yu Huang[1]
and Yuan-Zhong Wang[1]

[1]College of Traditional Chinese Medicine, Yunnan University of Chinese Medicine, Kunming 650500, People's Republic of China
[2]Institute of Medicinal Plants, Yunnan Academy of Agricultural Sciences, Kunming 650200, People's Republic of China

(iD) YW, 0000-0001-5376-757X

*Dendrobium* is the largest genus of orchids most of which have excellent medicinal properties. Fresh stems of some species have been consumed in daily life by Asians for thousands of years. However, there are differences in flavour and clinical efficacy among different species. Therefore, it is necessary for a detector to establish an effective and rapid method controlling botanical origins of these crude materials. In our study, three spectroscopies including mid-infrared (MIR) (transmission and reflection mode) and near-infrared (NIR) spectra were investigated for authentication of 12 *Dendrobium* species. Generally, two fusion strategies, reflection MIR and NIR spectra, were combined with three mathematical models (random forest, support vector machine with grid search (SVM-GS) and partial least-squares discrimination analysis (PLS-DA)) for discrimination analysis. In conclusion, a low-level fusion strategy comprising two spectra after pretreated by the second derivative and multiplicative scatter correction was recommended for discrimination analysis because of its excellent performance in three models. Compared with MIR spectra, NIR spectra were more responsible for the discrimination according to a bi-plot analysis of PLS-DA. Moreover, SVM-GS and PLS-DA were suitable for accurate discrimination (100% accuracy rates) of calibration and validation sets. The protocol combined with low-level fusion strategy and chemometrics provides a rapid and effective reference for control of botanical origins in crude *Dendrobium* materials.

# 1. Introduction

The relationship between health and diet has received more and more attention for high quality of life for many centuries. As a consequence, suitable diet supplements seem to be the best choice for the purpose. These diet supplements are the outcome of therapeutic experience of human beings in daily life over the generations. Especially herbal materials have become an important and essential portion in diet supplements of Asia (China, Korea, Indian and Japan), the USA, Canada, France, etc. [1].

*Dendrobium* is the largest genus of orchids containing more than 1000 species and 76 species (two variations) that are consumed as functional food and medical plants, especially in Yunnan and Zhejiang provinces of China [2]. Among these species, *Dendrobium officinale* is recommended as the functional food and medicinal material by China Food and Drugs Administration, while other three species (*D. nobile*, *D. chrysotoxum* and *D. fimbriatum*) just are regulated for clinical usage. Raw materials (fresh stems) of *D. officinale* are consumed as juice or directly chewed in China. What is more, one most special characteristic of the species is that there is nearly no residual after chewing. In addition, fresh stems are often twisted into the spiral shape similar to coiled spring because of their soft trait and high-content mucilage, for long-term storage and long-distance transport (electronic supplementary material, figure S1). After that, medicinal stems in commercial forms are consumed after processing as decoction or dishes with other food materials [3]. However, other congeneric species with the same properties are also processed into Fengdou form labelled as Tiepi Fengdou for high profits. Although non-Tiepi Fengdou crude materials also have been consumed for years, the cost for their cultivation is more inexpensive than that of *D. officinale*.

Chemical components and pharmacological research have indicated that *Dendrobium* species could produce polysaccharides, alkaloids, sesquiterpenoids, bibenzyl derivatives and other secondary metabolites. These constituents are responsible for anti-fatigue and anti-cancer activities, produce gastric ulcer protective effect, and regulate high standard of blood glucose, blood fat and blood pressure [4,5]. Polysaccharides are the main components in dry crude materials with immuno-modulatory, anti-tumour, anti-diabetic and antioxidant properties and so on [6,7]. However, the reviews mentioned above showed that these constituents and activities were species-dependent. For instance, bibenzyl combined with an intricately substituted oxygenic (benzo-) heterocycle is abundant in *D. officinale*, while coumarins existed in *D. densiflorum* and *D. thyrsiflorum* in higher content compared with other *Dendrobium* species [7]. The effects of polysaccharides of *D. officinale* showed obviously inhibitory performance on *Escherichia coli* strains, whereas *D. hookerianum* had the most remarkable antibacterial effect on *Bacillus subtilis* [8]. The differences in chemical components and pharmacological activities may be the deciding factors to choose a particular species. Therefore, quality analysts should establish an effective and rapid approach for investigation of botanical origins from different forms named—Fengdou.

Modern analytical techniques have provided several protocols for traceability of botanical origins in crude Fengdou materials. Among these analytical techniques, molecular phylogeny analysis was believed the most powerful method for species authentication. The method compensated for deficiencies of morphological distinction when some species without flowers was difficult to be traced their morphological classification [9,10]. High-performance liquid chromatography combined with mass spectra or other measurements is mainly applied for metabolic analysis to investigate the variation of chemical information caused by some factors such as harvesting period [11,12], species difference [13], geographical origins [14], etc. For research of content fluctuation of polysaccharides, saccharide mapping was recommended as an effective technique for quality assessment of different species [15,16]. In contrast with these methods, spectroscopy combined with chemometrics is a rapid approach, and it has been extensively applied for traceability of *Dendrobium* species [17]. Based on the minor absorbance difference of near-infrared (NIR) spectra, congeneric species could be successfully distinguished using principal component analysis and partial least-squares discrimination analysis (PLS-DA) [18]. In addition, mid-infrared (MIR) spectra and ultraviolet–visible (UV–Vis) spectra also have been reported for authentication of *Dendrobium* species [19–21] which will get confused with other species. Although these spectroscopies as a green and non-destructive method reflected a portion of metabolic information [22,23], the single spectroscopy was not enough for comprehensive assessment of *Dendrobium* species.

Multispectroscopy fusion (a part of data fusion) has gained attention in the discrimination of food and herbal materials [24–28]. These analytical techniques complemented each other because every analytical measurement displayed individual chemical information. In our previous study, the combination between MIR and UV–Vis spectra with the help of PLS-DA was successfully applied for authentication of 11 *Dendrobium* varieties [29]. However, the study did not contain the most popular species—*D. officinale* which was the main species in the herbal market. What is more, the fusion

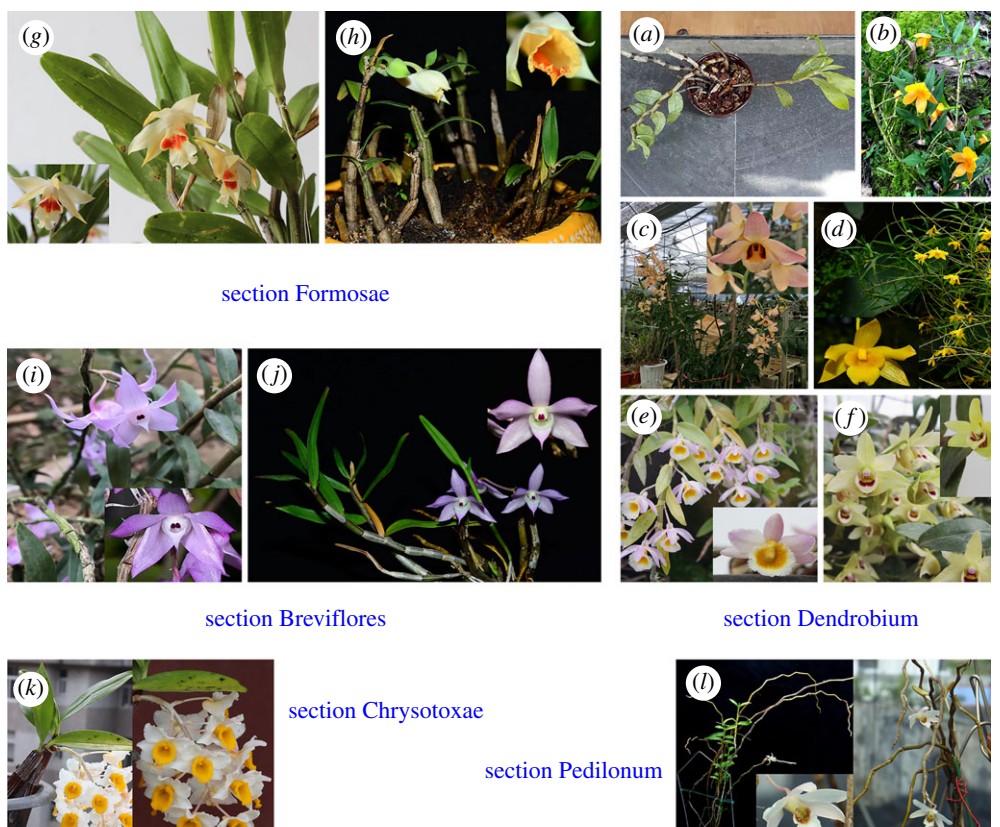

**Figure 1.** The morphological characteristics and section of each *Dendrobium* species as well as their flowers. (*a*) *Dendrobium hookerianum*, (*b*) *D. lohohense*, (*c*) *D. moschatum*, (*d*) *D. hancockii*, (*e*) *D. loddigesii*, (*f*) *D. officinale*, (*g*) *D. cariniferum*, (*h*) *D. longicornu*, (*i*) *D. aduncum*, (*j*) *D. hercoglossum*, (*k*) *D. thyrsiflorum* and (*l*) *D. xichouense*.

strategy with complementary NIR and MIR spectra has not been reported for authentication of the multi-origins of Fengdou crude materials.

Our main objective of the present study was to investigate the feasibility of combination between NIR and attenuated total reflectance Fourier infrared spectroscopy with the aid of three mathematical discrimination models. Besides original herbal materials (*D. officinale*) of Fengdou, other 11 easy-confusion species were also included for model establishment. Finally, the most optimal model was used for distinguishing *D. officinale* from other species. The well-established model was recommended as a reference for distinction of original plants of the crude materials applied in the functional food.

# 2. Material and methods

## 2.1. Plant materials and chemical reagent

Fresh stems of 12 *Dendrobium* species including 119 individuals were obtained from Wenshan University in Wenshan Prefecture, Yunnan Province of China. Plants' Latin name and section category were authenticated by Prof. H.-Y.H. according to their morphological characteristics of stems, leaves and inflorescences. Some species without inflorescences were transplanted into our greenhouse for blossom before authentication. Accurate Latin name and traditional use in national minorities of China are summarized in table 1. Because of the mature conservation condition of plant specimens, these original plants were deposited in Institute of Medicinal Plants, Yunnan Academy of Agricultural Sciences located in Kunming city in Yunnan Province. Original plant pictures of 12 *Dendrobium* species with amplified flowers are put together in figure 1, classified according to their section. Fresh stems of original plants were separated from the whole plants and dried at 45°C in the shade until constant weight. Dried stems without leaves of each sample were broken into powder with high-speed disintegrator (Tianjin Huaxin Instrument Factory, China). Sample powders were screened through an 80 stainless mesh sieve to obtain same-sized particles. The powders after sieving were

**Table 1.** Detained sample information including their application in daily life of China.

| code | species | section | commercial name | minorities | medical parts | medical purpose | references |
|---|---|---|---|---|---|---|---|
| 1 | *D. hookerianum* | section Dendrobium | — | — | — | — | — |
| 2 | *D. thyrsiflorum* | section Chrysotoxae | — | — | — | — | — |
| 3 | *D. cariniferum* | section Formosae | — | Lahu | fresh stems | external use for traumatic injury and bone injury; internal use for throat itching and cough | [30] |
| 4 | *D. hancockii* | section Dendrobium | Diaolanxi Fengdou | Zang | stems | fever and emesis caused by various factors | Zhongguozangyao |
| 5 | *D. aduncum* | section Breviflores | Shuicao Fengdou; Zipi Fengdou | Yi | whole | bone injury, swelling and pain | Ailao |
| 6 | *D. hercaglossum* | section Breviflores | Jizhualan | Lahu | fresh stems | external use for traumatic injury and bone injury; internal use for throat itching and cough | [30] |
| 7 | *D. longicornu* | section Formosae | — | — | — | — | — |
| 8 | *D. moschatum* | section Dendrobium | — | — | — | — | — |
| 9 | *D lohohense* | section Dendrobium | — | — | — | — | — |
| 10 | *D. xichouense* | section Pedilonum | — | — | — | — | — |
| 11 | *D. loddigesii* | section Dendrobium | Diaolan Fengdou; Xiaohuangcao Fengdou | Lahu | fresh stems | external use for traumatic injury and bone injury; internal use for throat itching and cough | [30] |
| 12 | *D. officinale* | section Dendrobium | Tiepi Fengdou; Xifengdou | Lahu | fresh stems | external use for traumatic injury and bone injury; internal use for throat itching and cough | [30] |

stored in dry and dark zip-lock bags for further two kinds of spectra scan. Potassium bromide (KBr) powder was purchased from Tianjin Fengchuan Fine Chemical Research Institute (Tianjin, China).

## 2.2. Near-infrared diffuse reflectance spectroscopy

An Antaris II spectrometer (Thermo Fisher Scientific, USA) with integrating sphere diffused reflection mode was used for obtaining NIR spectra of sample powders. Spectrum scan range was $10\,000$–$4000\,cm^{-1}$. Resolution was $8\,cm^{-1}$ and each sample power has 64 accumulated scans. The obtained spectra were recorded as the logarithm of reciprocal reflectance, log(1/reflectance). In order to reduce the operation error, 20.0 g powder was filled up a uniform glass vessel to scan one by one. In addition, humidity and room temperature were kept at 30% and 25°C, respectively, in the course spectra collection to exclude interference of laboratory air. Finally, the average spectrum from triplicate measurement was calculated as the final spectrum for mathematical analysis.

## 2.3. Fourier mid-infrared spectroscopy

The dry stem powder was analysed by Fourier transform mid-infrared spectrometer (FTIR) with deuterated triglycine sulfate (DTGS) detector equipped with attenuated total reflectance (ATR) accessory (Frontier Perkin Elmer). Each spectrum was scanned from 4000 to $650\,cm^{-1}$ at a resolution of $4\,cm^{-1}$ and 16 accumulations scans. What is more, three replicates were performed successively for eliminating operation error. The sample powder was put on the central of O-ring metal which was placed around the measuring unit. With the help of the pressure tower (Thermo Fisher Scientific micrometric pressure device), the sample powder was pressed tightly until $131 \pm 1$ bar on the scale of the micrometric pressure device to generate a constant layer thickness. After each measurement, the apex of the pressure tower and the surface of ATR crystal were carefully wiped with lint-free tissues containing deionized water. The next measurement was conducted after drying to avoid the interference between different species or different individuals from the same species. Before each measurement, the laboratory air spectrum (25°C and 30% RH) was recorded to check for remaining water and sample residues as well as background deduction. Three replicates were calculated for the average spectrum of each sample.

Besides the reflection mode analysis mentioned above, transmission FTIR spectroscopy was also applied based on direct powder KBr tablet method [31]. Briefly, the refined powder (1.2 mg) of each sample was sufficiently ground with KBr for obtaining a thin and hyaline tablet. The blank KBr tablet was scanned prior to each measurement for background deduction. The scan range was from 4000 to $400\,cm^{-1}$. Other measurement parameters and laboratory conditions were the same as those of the ATR reflection mode.

## 2.4. Spectra data pretreatment

The concentration of chemical components and physical properties are two main factors leading to minor spectral difference between two objects [32]. In addition, there are also overlapping absorbance peaks and noisy signals which would lead to interference for further discrimination purpose. The performance of the mathematical model using digital dataset often depends on how the original spectra are pretreated [33]. Therefore, a second derivative was used for increasing resolution and amplifying the weak absorbance peaks [34,35]. Multiplicative scatter correction was developed for correction of the variation caused by light scattering of sample particle size existing in these three measurements [36]. Two preprocessing methods and data transform were computed by SIMCA 14.1 software (Umetrics, Umeå, Sweden).

After selection between reflection and transmission mode of MIR spectroscopy, the optimal choice reflecting most of original spectra information was combined with NIR spectra for three supervised pattern recognition methods below. Moreover, two strategies were used for investigation of model performance.

Low-level fusion strategy: two kinds of spectra were directly connected with each other after pretreatment. The number of rows was the same as the number of samples, while the number of columns was equal to the total of variables information from two kinds of spectra. Prior to combination, NIR and FTIR datasets were normalized into $(-1, 1)$ for excluding the influence of the vertical coordinate.

Mid-level fusion strategy: NIR spectra and FTIR spectra were conducted with principal component analysis for dimensionality decline of high-dimensionality data. The principal components (PCs) of each spectra dataset which had linear correlation with original spectra matrices were extracted based

on eigenvalue greater than 1. Furthermore, these PCs were recombined with each other for chemometrics analysis. Therein, 20 PCs showing 90.83% variances from NIR spectra and 22 PCs interpreting 77.78% variable information from ATR-FTIR spectra were extracted for fusion analysis.

Prior to model establishment, two data matrices (low- and mid-fusion dataset) were divided into calibration set (two-thirds of total samples) and validation set (one-third of total samples) with the help of the Kennard–Stone algorithm in Matlab 2014a software (MathWorks, USA). The algorithm avoided the non-repeatability of random selection and evaluated whole measurement samples with the largest Euclidian distance [37,38]. The method bifurcating the dataset was beneficial for model discrimination ability and avoided unrepeatable results if the samples belonging to validation set were selected randomly.

## 2.5. Random forest

The published literature regarding random forest has shown successful applications for discrimination of geographical origins [26,28] and harvesting time [14] in crude herbal materials. However, there were no reports for species discrimination, especially *Dendrobium* species which are easily confused with other species. The unknown samples were predicted according to the well-established model which was an ensemble of trees (forest) where each tree voted for the most popular class [39]. The forest confirms the classification label of one sample which has the most votes number from the model calculation [32]. There are two crucial parameters that needed to be adjusted for low error and high classification performance. The initial parameter is the number of tree ($n_{tree}$). Based on the lowest out-of-band (OOB) error, the best $n_{tree}$ is used for selection of split variables ($m_{try}$). The first combination between $n_{tree}$ and $m_{try}$ is responsible for re-constructed matrix comprising important variables in the light of permutation accuracy importance. After that, the new matrix is computed similar to the above-mentioned two steps for ultimate model parameters. Of course, it will not need further parameter selection using importance variables matrix if the model shows excellent model performance using the original matrix. The accomplishment of random forest was calculated by randomForest R package (v. 3.6-12) [40].

## 2.6. Support vector machine

There is published literature for discrimination analysis for geographical traceability of herbal materials using a support vector machine with grid search (SVM-GS) [28,41,42]. It is not reported that SVM-GS is applied for classification of original species in the field of herbal materials. As a powerful classifier, the nonlinear classification model mainly depends on hyperplane and kernel function. Generally, the model has excellent generalization ability compared with other multivariate statistical approaches [43]. Original variable distribution is mapped into high-dimensional space where hyperplane plays an important role for detailed classification. Usually, two vital parameters, penalty parameter coast ($c$) and kernel parameter gamma ($g$), are responsible for the model performance. Grid search is the method for selection of two important parameters. In the process of calculation, each pair of ($c$, $g$) is independent in parallel with an accuracy rate of cross-validation. The best combination of the two parameters is based on the highest accuracy rate. Finally, those unknown samples belonging to validation set are verified with the well-establishment model. The calculation of the model was computed by the Matlab software.

## 2.7. Partial least-squares discrimination analysis

PLS-DA is a variant of partial least-squares regression. Different from regression analysis which is used for modelling relationship between dependent variables ($Y$) and independent variables ($X$), PLS-DA tries best to find a relationship between variables matrix ($X$) and label dataset ($Y$) for prediction of unknown samples. Therein, a fictitious $Y$ matrix comprised one and zero where an observation has the value 1 for specific class, while other classes are defined as code 0 [34]. The $X$ matrix consists of original data after pretreatment is transformed into some latent variables (LVs) which are linear correlations to original data. In our study, the first $x$ LVs were selected in the light of the maximum eigenvalue greater than 1. The model was established using calibration set with sevenfold cross-validation. Root mean square error of cross-validation (RMSECV), root mean square error of prediction (RMSEP), $R^2(X)$ and $Q^2(Y)$ were used for evaluation of the model performance. $R^2(X)$ shows the percentage of original dataset by model explanation, while $Q^2(Y)$ indicates prediction ability of the well-established model. Moreover, permutation test was calculated for validating the fitting degree of PLS-DA model according to results of $R^2$-intercept and $Q^2$-intercept. Permutation plot was used for assessing the risk that the current

discrimination model was spurious, such as the fact that the model just fitted the calibration set well but did not predict the relevant $Y$ category well for new observations. If $Q^2$ values of each category to the left were lower than the original points to the right and the intercepts of regression line of the $Q^2$-points were below zero, the model will be robust. In addition, all $R^2$ values to the left were lower than the original points to the right. The combination performance of $Q^2$ and $R^2$ (as well as $R^2$-intercept and $Q^2$-intercept) was the indication for the validity of discrimination model. The permutation test of each category model was conducted with 30 iterations. The discrimination model was computed by the SIMCA 14.1 software.

## 2.8. Evaluation of discrimination models

Besides accuracy rates of calibration and validation set, the evaluation of model performance was also provided by sensitivity, specificity, precious and efficiency. Three additional parameters were calculated based on four conceptions: true positive (TP), true negative (TN), false positive (FP) and false negative (FN). The four conceptions were used for further calculation of four parameters

$$\text{Sensitivity} = \frac{TP}{TP + FN},$$

$$\text{Specificity} = \frac{TN}{TN + FP},$$

$$\text{Precision} = \frac{TP}{TP + FP}$$

and

$$\text{Efficiency} = \sqrt{\text{Sensitivity} \times \text{Specificity}}.$$

Generally, four parameters were calculated synergistically for evaluation of model performance [44]. Among these parameters, sensitivity (true-positive rate) displays the fraction of samples belonging to a defined class which is correctly accepted by the category. The value of specificity is equal to the true-negative rate which indicates that samples do not belong to the specified class that is rejected the modelled class. Efficiency is a summarizing parameter for both sensitivity and specificity. Furthermore, precision is the ratio between accepted samples in specified samples and totally accepted samples of the model in calibration or validation set.

# 3. Results and discussion

## 3.1. Interpretation of near-infrared spectra

NIR spectra mainly reflect overtones and combinations of fundamental vibrations in absorbance hydrogen, such as C–H, N–H, O–H [45]. Figure 2 shows the original averaged NIR spectra of 12 *Dendrobium* species. As for the detailed spectra interpretation of these 12 species, our previous study has shown attribution of these peaks in detail [29].

For intuitive visualization of minor difference among these species, PCA was the first to be computed using spectra dataset after pretreatment which was based on the first two PCs that explained 43.04% variances of total spectra information. As figure 3*a* displays crude materials of functional food (*D. officinale*) could be clearly separated from other species. Except for *D. loddigesii*, *D. thyrsiflorum* and *D. hookerianum*, other eight species were clustered. To better reflect the difference between *D. officinale* and other species, figure 3*b* shows the cluster result using the score plot of PCA based on two categories.

## 3.2. Interpretation of mid-infrared spectra

Due to the fact that score plots (electronic supplementary material, figure S2A,B) did not show satisfying cluster performance using transmission FTIR spectra (electronic supplementary material, figure S3) after pretreatment by SD and MSC, ATR-FTIR spectra (figure 4) were recommended for further fusion analysis with NIR spectra matrix. Even though the first two PCs interpreted 36.34% original information which was more than that ATR-FTIR spectra (29.25%) (figure 5). In addition, there were similar spectra characteristics just in terms of spectra peaks between 4000 and 1000 cm$^{-1}$, which has the ability to reflect most of chemical information of these species.

Therein, averaged spectra of 12 *Dendrobium* species are displayed in figure 4. Among these spectral variables, the absorbance at 3350.70 cm$^{-1}$ (O–H stretching vibration) combined with band range between 1200 and 1030 cm$^{-1}$ is responsible for the dominant polysaccharide in these crude materials

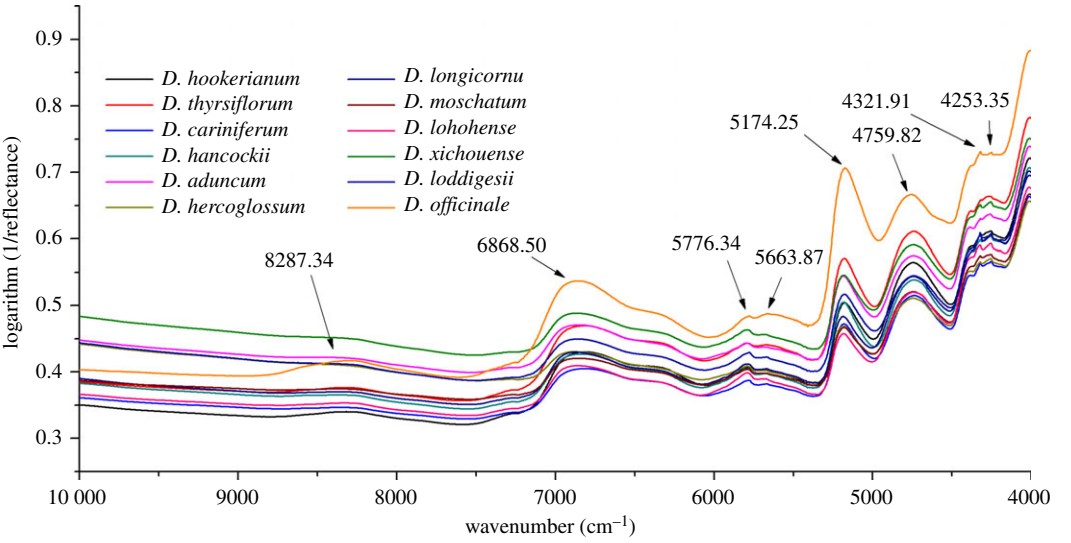

**Figure 2.** Average NIR spectra of 12 *Dendrobium* species.

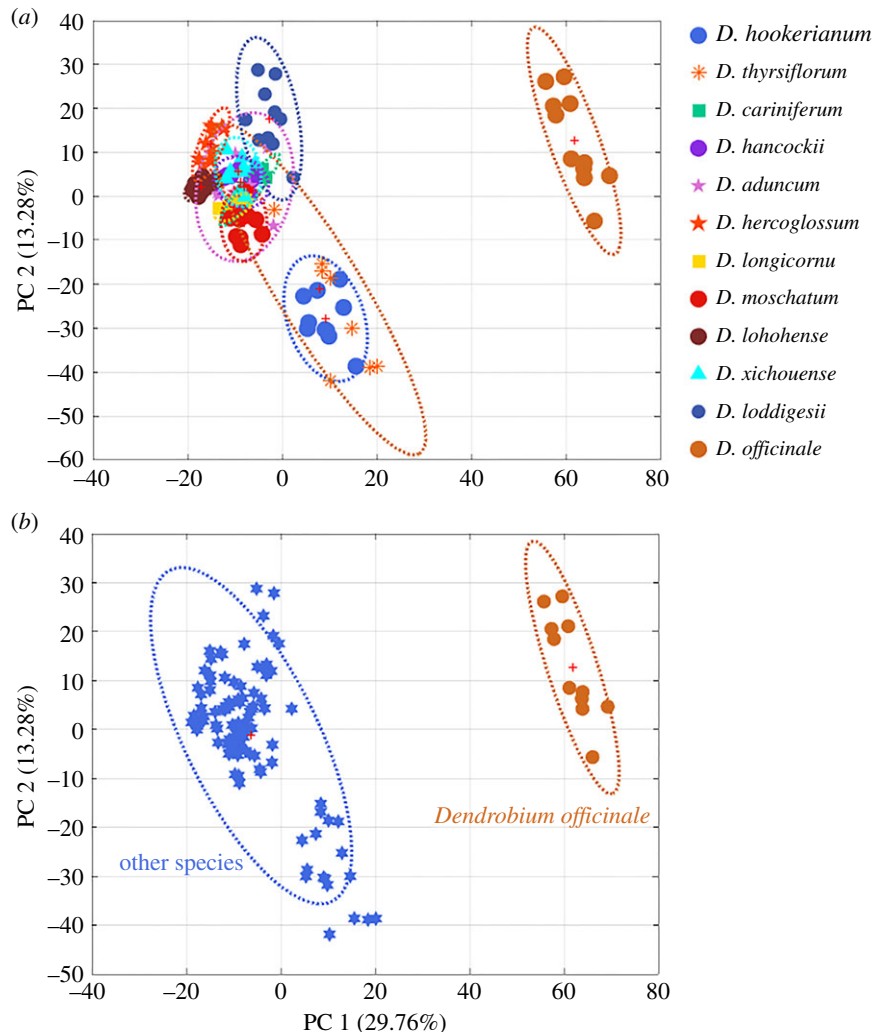

**Figure 3.** Score plots of PCA for 12 *Dendrobium* species using NIR spectra after pretreatment. (*a*) PCA containing 12 categories and (*b*) PCA containing two categories.

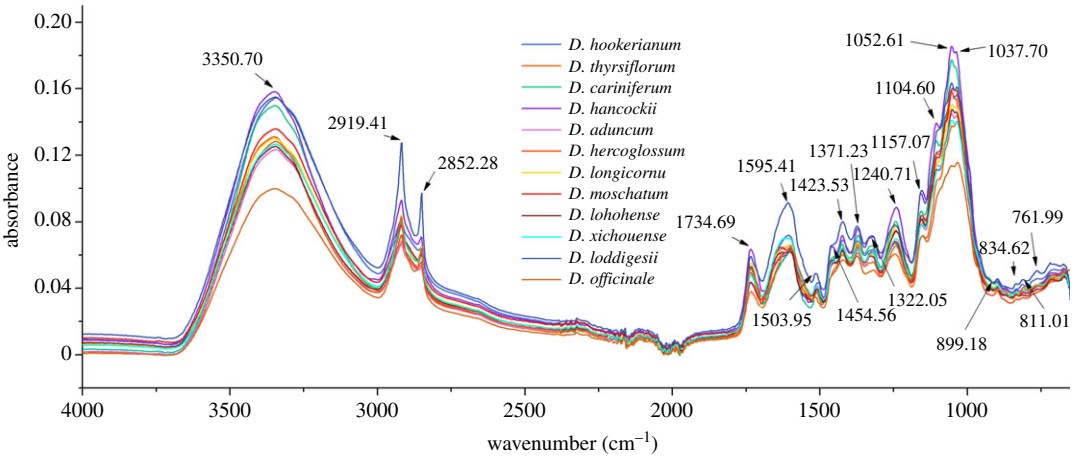

**Figure 4.** Average ATR-FTIR spectra of 12 *Dendrobium* species.

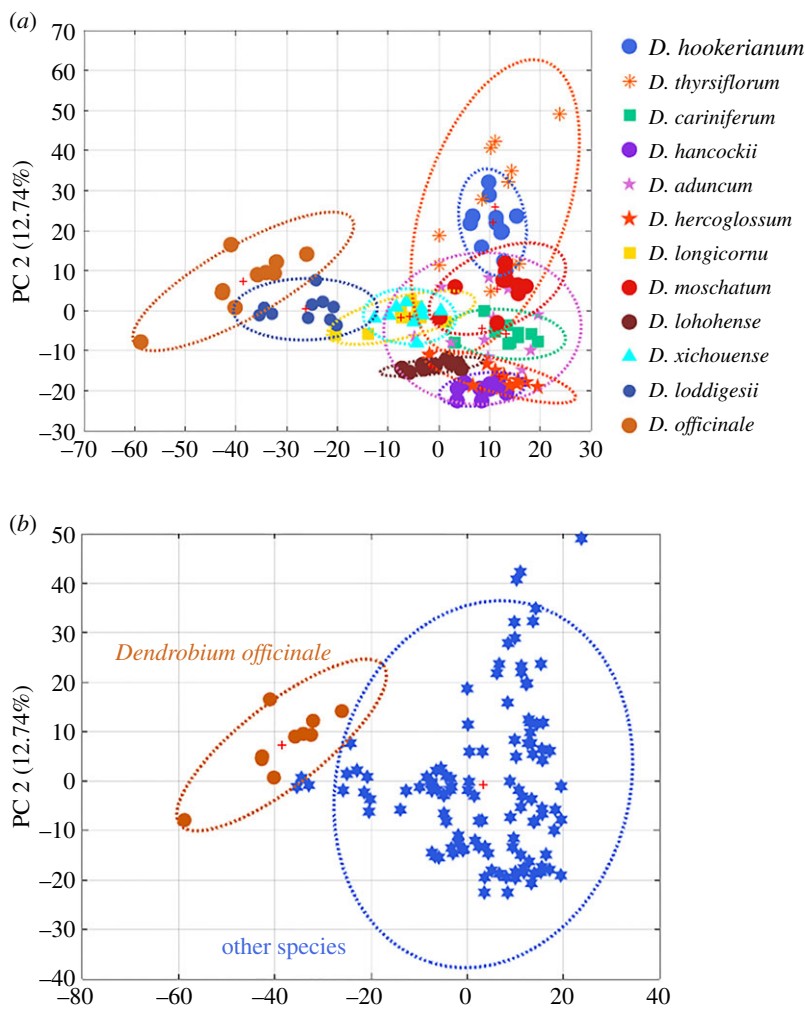

**Figure 5.** Score plots of PCA for 12 *Dendrobium* species using ATR-FTIR spectra after pretreatment. (*a*) PCA containing 12 categories and (*b*) PCA containing two categories.

from *Dendrobium* plants [5,46]. Absorbance peaks between 900 and 650 cm$^{-1}$ were the fingerprint range of polysaccharide which is the main active components within these herbs [7]. Herein, *Dendrobium* polysaccharides mainly contributed anti-tumour, anti-diabetic, immuno-modulatory and antioxidant properties. What is more, 2819.41 and 2852.28 cm$^{-1}$ are the main common absorbance of symmetrical

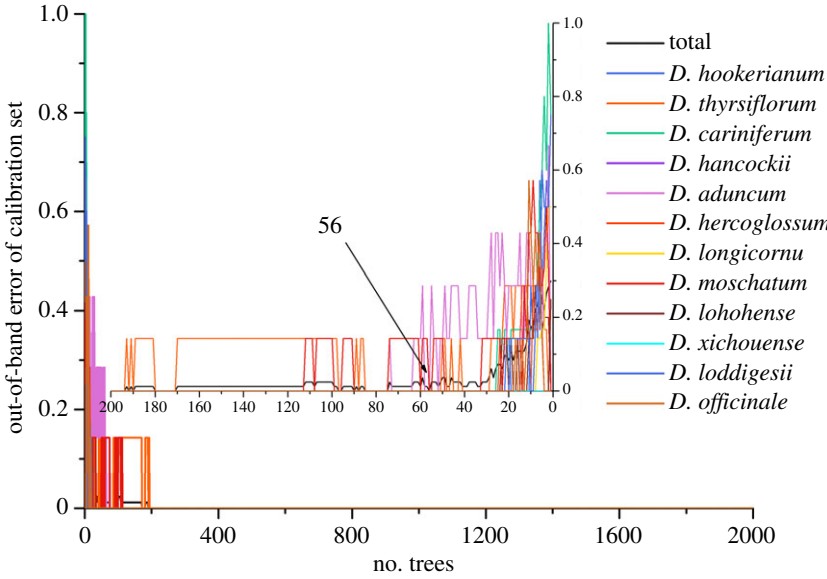

**Figure 6.** The selection results of $n_{tree}$ using the low-level fusion strategy.

and asymmetrical stretching vibration belonging to -CH$_3$ and -CH$_2$-, respectively, which exist in most of the chemical components. Other two peaks at 1423.53 and 1371.23 cm$^{-1}$ are bending vibration of -CH$_3$ and -CH$_2$-, respectively. Absorbance at 1734.69 cm$^{-1}$ is a classical C=O functional group. Absorbance between 1450 and 1200 cm$^{-1}$ mainly indicates the mixed band concerning lipoid, acid amide and benzene ring [19]. The published report shows that gigantol containing benzene ring has anti-inflammatory, antioxidant anti-tumoural, anti-cataract and anti-mutagenic effects [4]. Just in terms of steep peak at 1595.41 cm$^{-1}$, *D. loddigesii* which is the main original material of Xiaohuangcao Fengdou could be obviously discriminated from other species. At the absorbance around 1454.56 cm$^{-1}$, *D. hancockii* has special absorbance with two weak peaks in contrast with other species. Between mixed band at 1450 and 1200 cm$^{-1}$, *D. officinale* shows the lowest absorbance obviously.

## 3.3. Random forest

Generally, two steps are recommended for the calculation of random forest in which variable selection and parameter optimization are finished. It has been validated that the two recommended steps were beneficial for model performance in the published literature [14,26,28]. However, the model conclusion could not be reproduced in the present study.

It has been mentioned that two fusion strategies were applied for discrimination analysis. Initially, low-level fusion was worthy for selection of important variables because there were 3266 variables after the combination of two kinds of spectra. Before important variable selection based on permutation accuracy importance, two vital parameters were the first to select according to the lowest OOB error in calibration set. The results are displayed in figures 6 and 7 in which the best $n_{tree}$ was 56, while $m_{try}$ was 47. Discrimination results were obtained using the original matrix with two optimal parameters. According to the vote results (electronic supplementary material, table S1) in calibration set, the detailed classification of the calibration set is displayed in table 2, which indicated that nearly all of samples were correctly discriminated except two samples belonging to *D. thyrsiflorum* and *D. xichouense*. *Dendrobium thyrsiflorum* was equally discriminated as the defined species and *D. hookerianum*. *Dendrobium xichouense* was simultaneously voted for itself and *D. aduncum*. Then, two misclassified samples were defined as no class according to the votes. Additionally, four evaluation parameters were equal to 1 which showed an excellent model performance in precision and efficiency. Based on the well-establishment model, one-third of the unknown sample was inserted in the model. The results (table 3) indicated that all samples were correctly classified into their defined categories according to the vote results in electronic supplementary material, table S2. However, the model performance did not improve after variable selection. Electronic supplementary material, table S3 displays that the discrimination performance was extremely poor, that the accuracy of calibration set was 65.00%, while the accuracy was 97.59% before

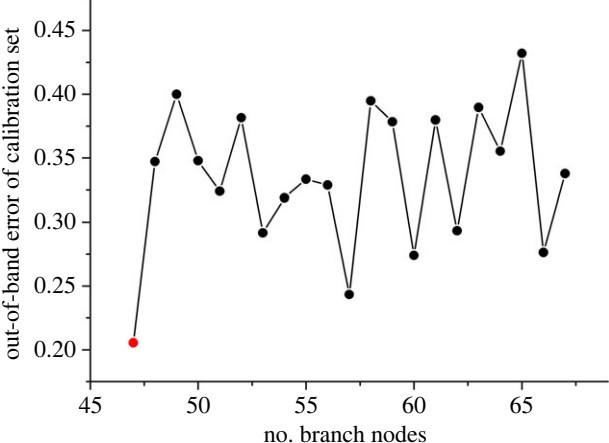

**Figure 7.** The selection results of $m_{try}$ using the low-level fusion strategy.

**Table 2.** Confusion matrix of random forest in calibration set using low-level fusion without important variable selection.

|  | 1 | 2 | 3 | 4 | 5 | 6 | 7 | 8 | 9 | 10 | 11 | 12 | no. class |
|---|---|---|---|---|---|---|---|---|---|---|---|---|---|
| 1 | 7 | 0 | 0 | 0 | 0 | 0 | 0 | 0 | 0 | 0 | 0 | 0 | 0 |
| 2 | 0 | 6 | 0 | 0 | 0 | 0 | 0 | 0 | 0 | 0 | 0 | 0 | 1 |
| 3 | 0 | 0 | 6 | 0 | 0 | 0 | 0 | 0 | 0 | 0 | 0 | 0 | 0 |
| 4 | 0 | 0 | 0 | 7 | 0 | 0 | 0 | 0 | 0 | 0 | 0 | 0 | 0 |
| 5 | 0 | 0 | 0 | 0 | 7 | 0 | 0 | 0 | 0 | 0 | 0 | 0 | 0 |
| 6 | 0 | 0 | 0 | 0 | 0 | 7 | 0 | 0 | 0 | 0 | 0 | 0 | 0 |
| 7 | 0 | 0 | 0 | 0 | 0 | 0 | 7 | 0 | 0 | 0 | 0 | 0 | 0 |
| 8 | 0 | 0 | 0 | 0 | 0 | 0 | 0 | 7 | 0 | 0 | 0 | 0 | 0 |
| 9 | 0 | 0 | 0 | 0 | 0 | 0 | 0 | 0 | 7 | 0 | 0 | 0 | 0 |
| 10 | 0 | 0 | 0 | 0 | 0 | 0 | 0 | 0 | 0 | 6 | 0 | 0 | 1 |
| 11 | 0 | 0 | 0 | 0 | 0 | 0 | 0 | 0 | 0 | 0 | 7 | 0 | 0 |
| 12 | 0 | 0 | 0 | 0 | 0 | 0 | 0 | 0 | 0 | 0 | 0 | 7 | 0 |
| sensitivity | 1 | 1 | 1 | 1 | 1 | 1 | 1 | 1 | 1 | 1 | 1 | 1 | — |
| specificity | 1 | 1 | 1 | 1 | 1 | 1 | 1 | 1 | 1 | 1 | 1 | 1 | — |
| precision | 1 | 1 | 1 | 1 | 1 | 1 | 1 | 1 | 1 | 1 | 1 | 1 | — |
| efficiency | 1 | 1 | 1 | 1 | 1 | 1 | 1 | 1 | 1 | 1 | 1 | 1 | — |

variable selection. The interesting conclusion could be interpreted that these contribution variables after selection were not enough for accurate discrimination analysis.

In addition, mid-level fusion strategy was also applied for the calculation by the discrimination model. It has been already mentioned that 42 PCs from two spectroscopies were combined for fusion analysis. Due to the fact that there were less extracted variables for computation of the discrimination analysis, the calculation was based on the initial analysis (without selection of important variables) mentioned in the low-level fusion analysis. In other words, the mid-level fusion dataset was directly used for two parameter selection according to variable selection. Based on the lowest OOB error in figures 8 and 9, the optimal number of trees ($n_{tree}$) is 229, while the node of branch ($m_{try}$) is 13. Furthermore, vote results in electronic supplementary material, table S4 were obtained after two vital parameter selections. Based on the vote results, the confusion matrix of calibration is displayed in table 4. The classification indicated that 98.80% of samples in calibration set were correctly discriminated into their defined classes except for one sample belonging to *D. officinale* which was incorrectly classified as *D. longicornu*. Although the performance of model with mid-level fusion

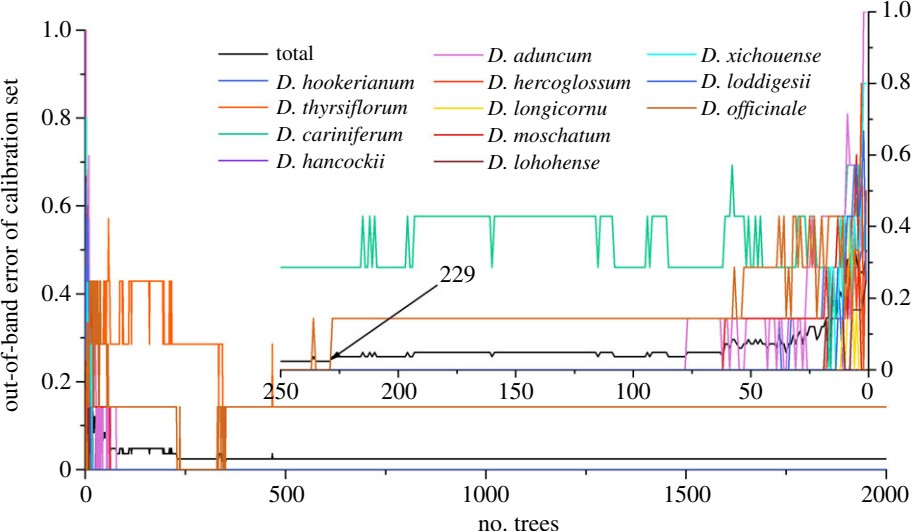

**Figure 8.** The selection results of $n_{\text{tree}}$ using the mid-level fusion strategy.

**Table 3.** Confusion matrix of random forest in validation set using low-level fusion without important variable selection.

| | 1 | 2 | 3 | 4 | 5 | 6 | 7 | 8 | 9 | 10 | 11 | 12 |
|---|---|---|---|---|---|---|---|---|---|---|---|---|
| 1 | 3 | 0 | 0 | 0 | 0 | 0 | 0 | 0 | 0 | 0 | 0 | 0 |
| 2 | 0 | 3 | 0 | 0 | 0 | 0 | 0 | 0 | 0 | 0 | 0 | 0 |
| 3 | 0 | 0 | 3 | 0 | 0 | 0 | 0 | 0 | 0 | 0 | 0 | 0 |
| 4 | 0 | 0 | 0 | 3 | 0 | 0 | 0 | 0 | 0 | 0 | 0 | 0 |
| 5 | 0 | 0 | 0 | 0 | 3 | 0 | 0 | 0 | 0 | 0 | 0 | 0 |
| 6 | 0 | 0 | 0 | 0 | 0 | 3 | 0 | 0 | 0 | 0 | 0 | 0 |
| 7 | 0 | 0 | 0 | 0 | 0 | 0 | 3 | 0 | 0 | 0 | 0 | 0 |
| 8 | 0 | 0 | 0 | 0 | 0 | 0 | 0 | 3 | 0 | 0 | 0 | 0 |
| 9 | 0 | 0 | 0 | 0 | 0 | 0 | 0 | 0 | 3 | 0 | 0 | 0 |
| 10 | 0 | 0 | 0 | 0 | 0 | 0 | 0 | 0 | 0 | 3 | 0 | 0 |
| 11 | 0 | 0 | 0 | 0 | 0 | 0 | 0 | 0 | 0 | 0 | 3 | 0 |
| 12 | 0 | 0 | 0 | 0 | 0 | 0 | 0 | 0 | 0 | 0 | 0 | 3 |
| sensitivity | 1 | 1 | 1 | 1 | 1 | 1 | 1 | 1 | 1 | 1 | 1 | 1 |
| specificity | 1 | 1 | 1 | 1 | 1 | 1 | 1 | 1 | 1 | 1 | 1 | 1 |
| precision | 1 | 1 | 1 | 1 | 1 | 1 | 1 | 1 | 1 | 1 | 1 | 1 |
| efficiency | 1 | 1 | 1 | 1 | 1 | 1 | 1 | 1 | 1 | 1 | 1 | 1 |

strategy was inferior to that of low-level fusion method just in terms of calibration set, the discrimination results of mid-level fusion are the same as those of low-level fusion not only by accuracy but also by four evaluation parameters in table 5.

## 3.4. Support vector machine

Different from the calculation of random forest, the computation of SVM-GS depended on two parameter selections ($c$ and $g$) rather than the combination between parameters ($n_{\text{tree}}$ and $m_{\text{try}}$) and important variables.

Initially, SVM-GS was calculated with low-level fusion dataset. In the light of the three-dimensional plots of parameters selection in figure 10, the discrimination performance is summarized in table 6, in

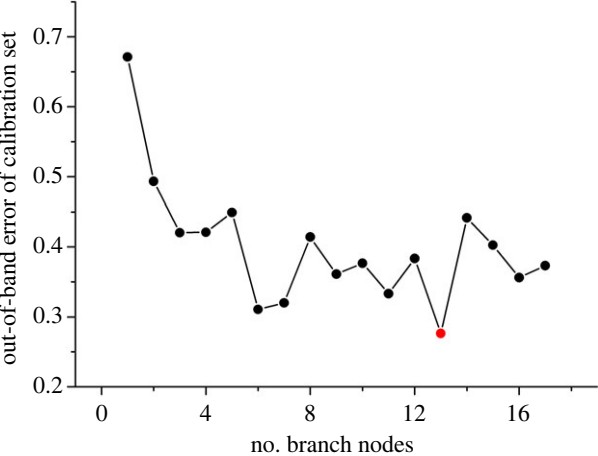

**Figure 9.** The selection results of $m_{try}$ using the mid-level fusion strategy.

**Table 4.** Confusion matrix of random forest in calibration set using mid-level fusion.

|  | 1 | 2 | 3 | 4 | 5 | 6 | 7 | 8 | 9 | 10 | 11 | 12 |
|---|---|---|---|---|---|---|---|---|---|---|---|---|
| 1 | 7 | 0 | 0 | 0 | 0 | 0 | 0 | 0 | 0 | 0 | 0 | 0 |
| 2 | 0 | 7 | 0 | 0 | 0 | 0 | 0 | 0 | 0 | 0 | 0 | 0 |
| 3 | 0 | 0 | 6 | 0 | 0 | 0 | 0 | 0 | 0 | 0 | 0 | 0 |
| 4 | 0 | 0 | 0 | 7 | 0 | 0 | 0 | 0 | 0 | 0 | 0 | 0 |
| 5 | 0 | 0 | 0 | 0 | 7 | 0 | 0 | 0 | 0 | 0 | 0 | 0 |
| 6 | 0 | 0 | 0 | 0 | 0 | 7 | 0 | 0 | 0 | 0 | 0 | 0 |
| 7 | 0 | 0 | 0 | 0 | 0 | 0 | 7 | 0 | 0 | 0 | 0 | 0 |
| 8 | 0 | 0 | 0 | 0 | 0 | 0 | 0 | 7 | 0 | 0 | 0 | 0 |
| 9 | 0 | 0 | 0 | 0 | 0 | 0 | 0 | 0 | 7 | 0 | 0 | 0 |
| 10 | 0 | 0 | 0 | 0 | 0 | 0 | 0 | 0 | 0 | 7 | 0 | 0 |
| 11 | 0 | 0 | 0 | 0 | 0 | 0 | 0 | 0 | 0 | 0 | 7 | 0 |
| 12 | 0 | 0 | 0 | 0 | 0 | 0 | 1 | 0 | 0 | 0 | 0 | 6 |
| sensitivity | 1 | 1 | 1 | 1 | 1 | 1 | 1 | 1 | 1 | 1 | 1 | 0.8571 |
| specificity | 1 | 1 | 1 | 1 | 1 | 1 | 0.9868 | 1 | 1 | 1 | 1 | 1 |
| precision | 1 | 1 | 1 | 1 | 1 | 1 | 0.8750 | 1 | 1 | 1 | 1 | 1 |
| efficiency | 1 | 1 | 1 | 1 | 1 | 1 | 1 | 1 | 1 | 1 | 1 | 0.9258 |

which samples belonging to calibration set and validation set were correctly discriminated as defined categories perfectly with $c = 0.3536$ and $g = 0.0039$. Furthermore, mid-level fusion protocol was also executed by SVM-GS. As the results of parameter optimization indicated in figure 11, excellent discrimination results (table 6) were obtained with $c = 2.0000$ and $g = 0.0652$ that both accuracy rates of calibration and validation sets were 100%.

In general, low-level fusion strategy seemed to be the best choice for species discrimination of 12 *Dendrobium* species when SVM-GS was applied for authentication, because the fusion strategy has low $c$ value. The published literature has interpreted that the lowest $c$ value meant robustness of the model [43].

## 3.5. Partial least-squares discrimination analysis

Different from the calculation of two models mentioned above, PLS-DA relies on more parameter selections. Model robustness and fitting are determined by these parameters which direct to the adjustment of dataset,

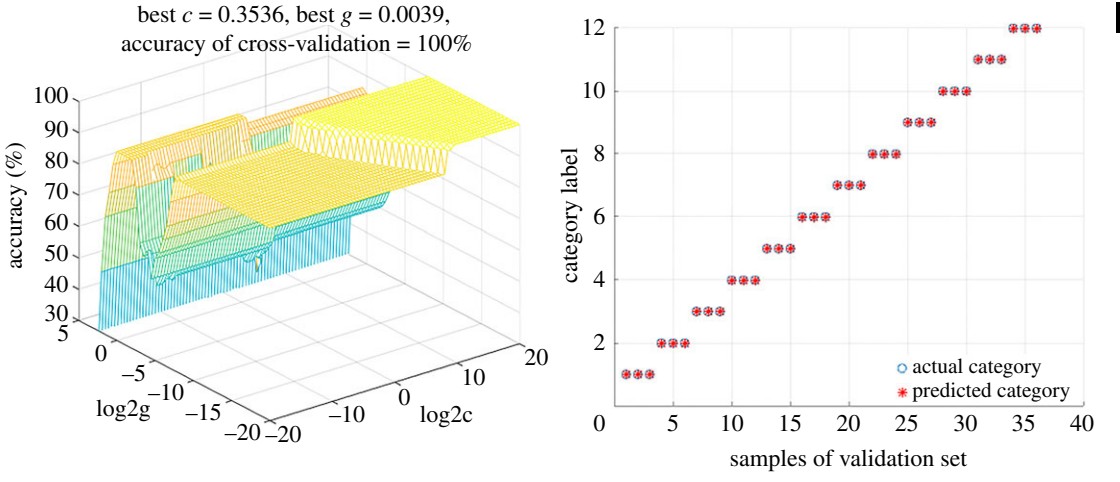

**Figure 10.** Results of SVM-GS method with the low-level fusion strategy.

**Table 5.** Confusion matrix of random forest in validation set using mid-level fusion.

|  | 1 | 2 | 3 | 4 | 5 | 6 | 7 | 8 | 9 | 10 | 11 | 12 |
|---|---|---|---|---|---|---|---|---|---|---|---|---|
| 1 | 3 | 0 | 0 | 0 | 0 | 0 | 0 | 0 | 0 | 0 | 0 | 0 |
| 2 | 0 | 3 | 0 | 0 | 0 | 0 | 0 | 0 | 0 | 0 | 0 | 0 |
| 3 | 0 | 0 | 3 | 0 | 0 | 0 | 0 | 0 | 0 | 0 | 0 | 0 |
| 4 | 0 | 0 | 0 | 3 | 0 | 0 | 0 | 0 | 0 | 0 | 0 | 0 |
| 5 | 0 | 0 | 0 | 0 | 3 | 0 | 0 | 0 | 0 | 0 | 0 | 0 |
| 6 | 0 | 0 | 0 | 0 | 0 | 3 | 0 | 0 | 0 | 0 | 0 | 0 |
| 7 | 0 | 0 | 0 | 0 | 0 | 0 | 3 | 0 | 0 | 0 | 0 | 0 |
| 8 | 0 | 0 | 0 | 0 | 0 | 0 | 0 | 3 | 0 | 0 | 0 | 0 |
| 9 | 0 | 0 | 0 | 0 | 0 | 0 | 0 | 0 | 3 | 0 | 0 | 0 |
| 10 | 0 | 0 | 0 | 0 | 0 | 0 | 0 | 0 | 0 | 3 | 0 | 0 |
| 11 | 0 | 0 | 0 | 0 | 0 | 0 | 0 | 0 | 0 | 0 | 3 | 0 |
| 12 | 0 | 0 | 0 | 0 | 0 | 0 | 0 | 0 | 0 | 0 | 0 | 3 |
| sensitivity | 1 | 1 | 1 | 1 | 1 | 1 | 1 | 1 | 1 | 1 | 1 | 1 |
| specificity | 1 | 1 | 1 | 1 | 1 | 1 | 1 | 1 | 1 | 1 | 1 | 1 |
| precision | 1 | 1 | 1 | 1 | 1 | 1 | 1 | 1 | 1 | 1 | 1 | 1 |
| efficiency | 1 | 1 | 1 | 1 | 1 | 1 | 1 | 1 | 1 | 1 | 1 | 1 |

**Table 6.** Results of GS-SVM method with mid-level fusion strategy.

|  | best $c$ | best $g$ | accuracy of calibration (%) | accuracy of validation (%) |
|---|---|---|---|---|
| low-level fusion | 0.3536 | 0.0039 | 100 | 100 |
| mid-level fusion | 2 | 0.0625 | 100 | 100 |

such as the complementary variable information [29] and pretreatment [47]. Our present study focused on multispectroscopy information fusion for investigation of the model performance.

Similar to the calculation of algorithms mentioned above, low-level fusion strategy was the first to be investigated by PLS-DA. Therein, 21 LVs explained for 79.10% spectra information, including ATR-FTIR

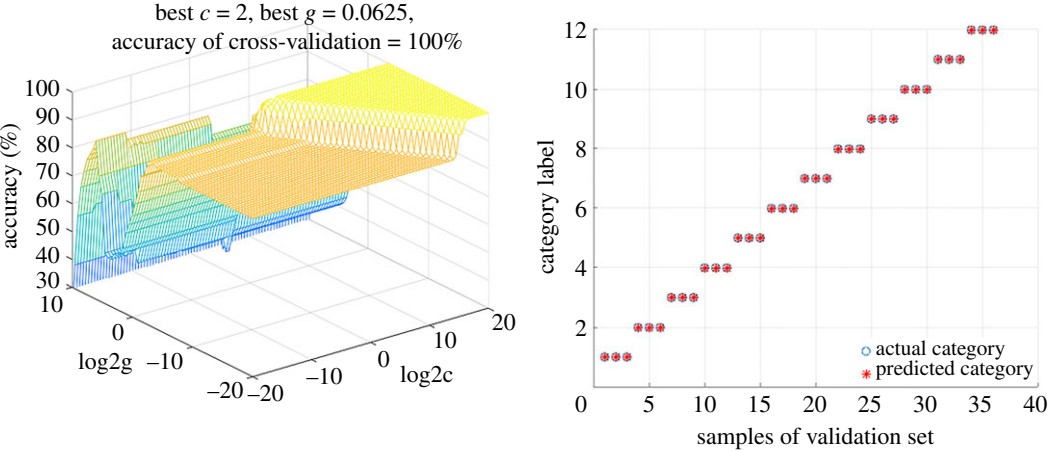

**Figure 11.** Results of SVM-GS method with the mid-level fusion strategy.

and NIR spectra after pretreatment, were used for model establishment. $Q^2$ value equal to 0.8667 indicated a strong prediction ability for unknown samples. Results of parameter calculation and discrimination results are displayed in table 7. These results indicated that not only calibration set but also validation set have an excellent discrimination ability with 100% discrimination accuracy and low error that both RMSECV and RMSEP were lower than 0.2. In addition, the permutation test with an iteration of 30 times showed that the model using low-level fusion strategy was robust because $R^2$-intercepts were more than 0.630 while $Q^2$-intercepts were lower than $-1.39$. Moreover, the absolute value of two intercepts was obviously lower than the $R^2$ and $Q^2$ values in the right of the corner of result plot of permutation test. For better evaluation of the model performance, four parameters of validation set instead of calibration set are calculated and given in table 7, in which these results of parameter calculation indicated perfect model precision and efficiency. What is more, both sensitivity and specificity were equal to 1, which showed that the model had strong ability to discriminate not only true positive but also true negative.

Besides low-level fusion strategy, mid-level fusion strategy was also used for PLS-DA model establishment. After a combination of 42 PCs from two kinds of spectra, 19 LVs were selected for the final model forming with 75.90% variable information and $Q^2 = 0.704$ (low prediction ability in contrast with low-fusion strategy). Compared with accuracy rates of low-level fusion in validation set, accuracy rates of mid-level fusion were inferior to those of low-level fusion strategy because one sample belonging to *D. officinal* category was misclassified into *D. xichouense* class. The misclassified sample showed poor parameter values in terms of specificity and efficiency in two categories. In addition, two error values also indicated that those of mid-level fusion strategy were higher than those of low-level fusion. The highest value of RMSECV was 0.2349 and the highest value of RMSEP was 0.1732; two values were higher than the error value of PLS-DA while using low-level fusion dataset. Except the two classes containing the misclassification individual, other samples belonging to other 10 species were correctly discriminated into defined categories in calibration and validation sets. In addition, the permutation test with iteration of 30 iterations was also used for investigation of robust and fitting in the model which was the same as the result using low-level fusion strategy. The results of iteration calculation and other parameters are displayed in table 8.

Based on the mid-level fusion strategy, a bi-plot (figure 12) was further applied for investigation of the contribution of two spectra in the classification. Generally, the nearest PC explained that portion of spectral information contributed for the discrimination of the species. For the detailed interpretation of these contributions, the nearest PC and its accumulation contribution as well as the spectra type are summarized in table 9. The results indicated that NIR spectra contributed for the discrimination of *D. hookerianum*, *D. cariniferum*, *D. hancockii*, *D. hercoglossum*, *D. moschatum*, *D. xichouense*, *D. loddigesii* and *D. officinale*. In contrast with the contribution of NIR spectra, ATR-FTIR spectra were responsible for the discrimination of *D. thyrsiflorum*, *D. aduncum*, *D. longicornu* and *D. lohohense*. Of course, these PCs were the main variables and gave less chemical information, and other PCs were also synergetic for the whole discrimination analysis of 12 *Dendrobium* species. In conclusion, the low-level fusion strategy was recommended for discrimination analysis of these 12 *Dendrobium* species because of model performance and classification accuracy.

**Table 7.** Results of PLS-DA method with low-level fusion strategy. Note: RMSECV, root mean square error of cross-validation; RMSEP, root mean square error of prediction; CS, calibration set; VS, validation set.

|  | 1 | 2 | 3 | 4 | 5 | 6 | 7 | 8 | 9 | 10 | 11 | 12 |
|---|---|---|---|---|---|---|---|---|---|---|---|---|
| RMSECV | 0.1108 | 0.1325 | 0.1042 | 0.0901 | 0.1960 | 0.1471 | 0.1166 | 0.1147 | 0.1094 | 0.1350 | 0.0884 | 0.0619 |
| RMSEP | 0.0950 | 0.0976 | 0.0571 | 0.0648 | 0.1238 | 0.1173 | 0.0866 | 0.0999 | 0.0812 | 0.1000 | 0.0650 | 0.0420 |
| $R^2$-intercepts | 0.675 | 0.649 | 0.648 | 0.630 | 0.643 | 0.643 | 0.656 | 0.668 | 0.655 | 0.648 | 0.633 | 0.645 |
| $Q^2$-intercepts | −1.26 | −1.30 | −1.39 | −1.36 | −1.38 | −1.36 | −1.26 | −1.33 | −1.29 | −1.33 | −1.18 | −1.29 |
| $R^2$ | 0.9804 | 0.9707 | 0.9732 | 0.9715 | 0.9622 | 0.9701 | 0.9570 | 0.9724 | 0.9671 | 0.9690 | 0.9735 | 0.9938 |
| $Q^2$ | 0.9029 | 0.8410 | 0.8995 | 0.9236 | 0.6057 | 0.7175 | 0.8629 | 0.8641 | 0.8675 | 0.8331 | 0.9119 | 0.9830 |
| accuracy of CS | 1 | 1 | 1 | 1 | 1 | 1 | 1 | 1 | 1 | 1 | 1 | 1 |
| accuracy of VS | 1 | 1 | 1 | 1 | 1 | 1 | 1 | 1 | 1 | 1 | 1 | 1 |
| sensitivity | 1 | 1 | 1 | 1 | 1 | 1 | 1 | 1 | 1 | 1 | 1 | 1 |
| specificity | 1 | 1 | 1 | 1 | 1 | 1 | 1 | 1 | 1 | 1 | 1 | 1 |
| precision | 1 | 1 | 1 | 1 | 1 | 1 | 1 | 1 | 1 | 1 | 1 | 1 |
| efficiency | 1 | 1 | 1 | 1 | 1 | 1 | 1 | 1 | 1 | 1 | 1 | 1 |

**Table 8.** Results of PLS-DA method with mid-level fusion strategy. Note: RMSECV, root mean square error of cross-validation; RMSEP, root mean square error of prediction; CS, calibration set; VS, validation set.

| | 1 | 2 | 3 | 4 | 5 | 6 | 7 | 8 | 9 | 10 | 11 | 12 |
|---|---|---|---|---|---|---|---|---|---|---|---|---|
| RMSECV | 0.1700 | 0.1993 | 0.2015 | 0.1801 | 0.2225 | 0.2349 | 0.1587 | 0.2232 | 0.1478 | 0.1782 | 0.2077 | 0.2250 |
| RMSEP | 0.1003 | 0.1058 | 0.0597 | 0.0799 | 0.0966 | 0.1167 | 0.0849 | 0.0823 | 0.1016 | 0.1449 | 0.0752 | 0.1732 |
| $R^2$-intercepts | 0.293 | 0.315 | 0.308 | 0.315 | 0.301 | 0.332 | 0.309 | 0.362 | 0.332 | 0.326 | 0.338 | 0.326 |
| $Q^2$-intercepts | −2.35 | −2.04 | −2.09 | −2.10 | 2.15 | −2.08 | −2.30 | −2.19 | −2.30 | −2.28 | −2.36 | −1.95 |
| $R^2$ | 0.9527 | 0.9562 | 0.9504 | 0.9503 | 0.8875 | 0.7929 | 0.9191 | 0.9000 | 0.8901 | 0.8787 | 0.9416 | 0.9812 |
| $Q^2$ | 0.7435 | 0.7230 | 0.8124 | 0.7772 | 0.6839 | 0.4246 | 0.8021 | 0.5529 | 0.8297 | 0.6605 | 0.7875 | 0.3815 |
| accuracy of CS | 1 | 1 | 1 | 1 | 1 | 1 | 1 | 1 | 1 | 1 | 1 | 1 |
| accuracy of VS | 1 | 1 | 1 | 1 | 1 | 1 | 1 | 1 | 1 | 1 | 1 | 0.6667 |
| sensitivity | 1 | 1 | 1 | 1 | 1 | 1 | 1 | 1 | 1 | 1 | 1 | 0.6667 |
| specificity | 1 | 1 | 1 | 1 | 1 | 1 | 1 | 1 | 1 | 0.9706 | 1 | 1 |
| precision | 1 | 1 | 1 | 1 | 1 | 1 | 1 | 1 | 1 | 0.7500 | 1 | 1 |
| efficiency | 1 | 1 | 1 | 1 | 1 | 1 | 1 | 1 | 1 | 0.9852 | 1 | 0.8165 |

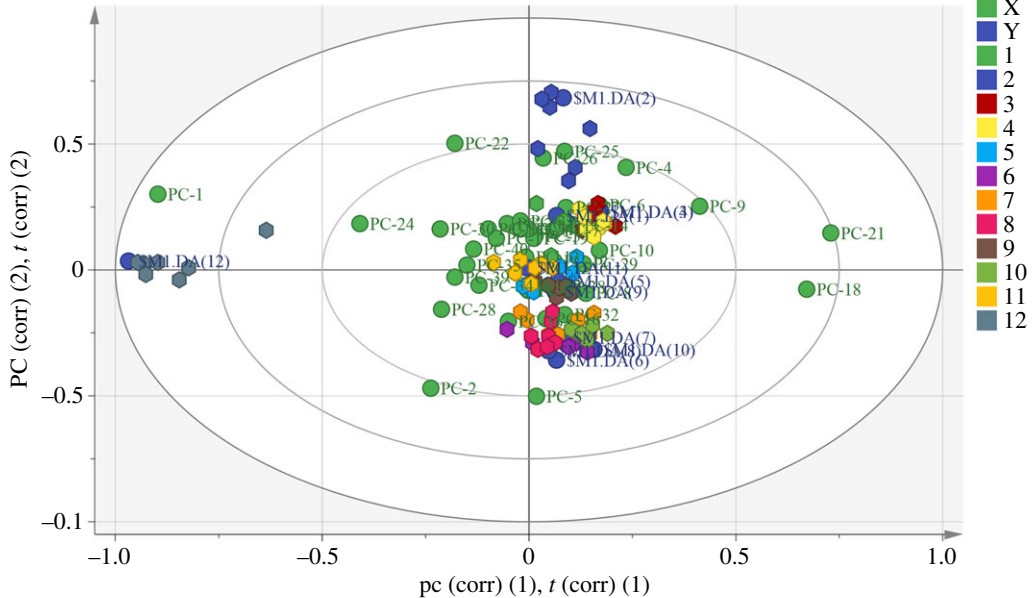

**Figure 12.** Bi-plot of PLS-DA with the mid-level fusion strategy.

**Table 9.** Results of bi-plot in PLS-DA with mid-level fusion strategy.

| species code | 1 | 2 | 3 | 4 |
|---|---|---|---|---|
| species | *D. hookerianum* | *D. thyrsiflorum* | *D. cariniferum* | *D. hancockii* |
| contribution PC | 7 | 25 | 6 | 6 |
| spectra information | NIR | ATR-FTIR | NIR | NIR |
| percentage of spectra information | 3.26% | 4.70% | 3.62% | 3.62% |
| species code | 5 | 6 | 7 | 8 |
| species | *D. aduncum* | *D. hercoglossum* | *D. longicornu* | *D. moschatum* |
| contribution PC | 38 | 16 | 32 | 16 |
| spectra information | ATR-FTIR | NIR | ATR-FTIR | NIR |
| percentage of spectra information | 0.96% | 1.29% | 2.10% | 1.29% |
| species code | 9 | 10 | 11 | 12 |
| species | *D. lohohense* | *D. xichouense* | *D. loddigesii* | *D. officinale* |
| contribution PC | 39 | 8 | 13 | 1 |
| spectra information | ATR-FTIR | NIR | NIR | NIR |
| percentage of spectra information | 0.97% | 2.94% | 1.81% | 29.76% |

## 3.6. Comparison of discrimination performance in three models

Based on the results from three models and two fusion strategies given above, the low-level fusion method was recommended as the optimal fusion method for the discrimination analysis of 12 *Dendrobium* species. It was interpreted that the low-level fusion strategy contains the most original information from different spectra which would reflect the maximum difference among 12 species. As for mid-level fusion method, the accuracy rate of calibration set in RF and accuracy rate of validation set in PLS-DA were less than 100%. The $C$ value of cross-validation in mid-level fusion was higher than that of low-level fusion in SVM. The poor performance of mid-level fusion could be explained that the variable transform might lose some chemical variable information, although it has rapid calculation speed.

# 4. Conclusion

In the present study, MIR spectra (reflection and transmission modes) and NIR spectra after pretreated by SD and MSC were investigated for species discrimination. Finally, ATR-FTIR (reflection mode) and NIR spectra were selected for two fusion strategies (low and mid level). In addition, three chemometrics models including RF, SVM-GS and PLS-DA were used for model establishment with two fusion matrices. Generally, the low-level fusion strategy with the aid of PLS-DA and SVM-GS was recommended as the final protocol for authentication of these 12 *Dendrobium* species, especially the functional materials: *D. officinale*.

Data accessibility. The datasets supporting this article have been uploaded as part of the electronic supplementary material.

Authors' contributions. Y.W. and Y.-Z.W. planned the research and wrote the manuscript. Z.-T.Z. and H.-Y.H. performed all the experiments and analyses.

Competing interests. We declare we have no competing interests.

Funding. This work was supported by the Key Project of Yunnan Provincial Natural Science Foundation (grant no. 2017FA049).

Acknowledgements. We thank Fei Liu (Normal University of Yuxi, Yuxi, China) for his photos of our original plants.

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
