## [Reviewer comments · Royal Society Open Science]

Review History

RSOS-182122.R0 (Original submission)

Review form: Reviewer 1

Is the manuscript scientifically sound in its present form?

No

Are the interpretations and conclusions justified by the results?

No

Is the language acceptable?

Yes

Is it clear how to access all supporting data?

Yes

Do you have any ethical concerns with this paper?

No

Have you any concerns about statistical analyses in this paper?

No

Recommendation?

Major revision is needed (please make suggestions in comments)

Comments to the Author(s)

The manuscript entitled "Original Plant Traceability of Dendrobium Species Using Multi-Spectroscopy Fusion and Mathematical Model" mainly presented an approach for fast identification of botanical origins of Dendrobium materials. Mid-infrared and NIR techniques combined with three different chemometric methods were used. Although lots of work has been done by the authors, the sections of results and discussion should be further improved. After careful review, I recommended major revision based on following points:

- (1) As stated by authors, MIR and UV-Vis spectra was used to authenticate the multi-origin Fengdou crude materials in Ref. [29]. What's the difference between these two articles? Just two different techniques?
- (2) Line 56, Page 6. The reason for accumulation number was wrong.
- (3) Why did authors choose to use these three methods?
- (4) The section of discussion was simple, and the advantages and disadvantages of these three methods should be discussed.
- (5) The sample number used for was limited. I wonder the reliability of the results.
- (6) How about the results of cross-validation for these three methods?
- (7) Grid search support vector machine should be replaced by support vector machine.
- (8) Line 60, Page 7. The authors stated reflection and transmission mode of mid-infrared spectroscopy were used in this experiments. However, the comparison for these two modes was not presented.
- (9) Language should be further improved. Lots of grammatical mistakes, such as Line 21-23, Page 6; Line 41-44, Page 7; Line 50-51, Page 7; Line 56-57, Page 7.

Review form: Reviewer 2

Is the manuscript scientifically sound in its present form?

No

Are the interpretations and conclusions justified by the results?

Yes

Is the language acceptable?

No

Is it clear how to access all supporting data?

Yes

Do you have any ethical concerns with this paper?

No

Have you any concerns about statistical analyses in this paper?

I do not feel qualified to assess the statistics

Recommendation?

Major revision is needed (please make suggestions in comments)

Comments to the Author(s)

The manuscript includes relevant information and applied techniques for the discrimination of *Dendrobium* species. However, it is poorly written which makes it hard to follow and comprehend. It is highly recommended to be revised and rewritten by a native speaker.

A discussion of the relevant bioactive components and their relation to the finding in the IR data is required. How do the species relate to each other with relevance to the metabolites is also important. In page 4 you mention that the species are quite different in their metabolites so how can you apply this to your discrimination results? The interpretation in page 12 deals with functional groups rather than prediction of actual metabolites.

Decision letter (RSOS-182122.R0)

01-Mar-2019

Dear Dr Wang:

Manuscript ID: RSOS-182122

Title: "Original Plant Traceability of *Dendrobium* Species Using Multi-Spectroscopy Fusion and Mathematical Model"

Thank you for submitting the above manuscript to Royal Society Open Science. Your paper was sent to reviewers and their comments are included at the bottom of this letter.

In view of the concerns raised by the reviewers, the manuscript has been rejected in its current form. However, a new manuscript may be submitted which takes into consideration these comments.

Please note that resubmitting your manuscript does not guarantee eventual acceptance, and that your resubmission will be subject to peer review before a decision is made.

Your resubmitted manuscript should be submitted by 29-Aug-2019. If you are unable to submit by this date please contact the Editorial Office.

On behalf of the Subject Editor Professor Anthony Stace and the Associate Editor Professor Hazel Cox

REVIEWER(S) REPORTS:

Associate Editor Comments to Author ():

RSC Associate Editor:

Comments to the Author:

(There are no comments.)

RSC Subject Editor:

Comments to the Author:

(There are no comments.)

Reviewers' Comments to Author:

Reviewer: 1

Comments to the Author(s)

The manuscript entitled "Original Plant Traceability of Dendrobium Species Using Multi-Spectroscopy Fusion and Mathematical Model" mainly presented an approach for fast identification of botanical origins of Dendrobium materials. Mid-infrared and NIR techniques combined with three different chemometric methods were used. Although lots of work has been done by the authors, the sections of results and discussion should be further improved. After careful review, I recommended major revision based on following points:

- (1) As stated by authors, MIR and UV-Vis spectra was used to authenticate the multi-origin Fengdou crude materials in Ref. [29]. What's the difference between these two articles? Just two different techniques?
- (2) Line 56, Page 6. The reason for accumulation number was wrong.
- (3) Why did authors choose to use these three methods?
- (4) The section of discussion was simple, and the advantages and disadvantages of these three methods should be discussed.
- (5) The sample number used for was limited. I wonder the reliability of the results.
- (6) How about the results of cross-validation for these three methods?
- (7) Grid search support vector machine should be replaced by support vector machine.
- (8) Line 60, Page 7. The authors stated reflection and transmission mode of mid-infrared spectroscopy were used in this experiments. However, the comparison for these two modes was not presented.
- (9) Language should be further improved. Lots of grammatical mistakes, such as Line 21-23, Page 6; Line 41-44, Page 7; Line 50-51, Page 7; Line 56-57, Page 7.

Reviewer: 2

Comments to the Author(s)

The manuscript includes relevant information and applied techniques for the discrimination of *Dendrobium* species. However, it is poorly written which makes it hard to follow and comprehend. It is highly recommended to be revised and rewritten by a native speaker.

A discussion of the relevant bioactive components and their relation to the finding in the IR data is required. How do the species relate to each other with relevance to the metabolites is also important. In page 4 you mention that the species are quite different in their metabolites so how can you apply this to your discrimination results? The interpretation in page 12 deals with functional groups rather than prediction of actual metabolites.

Author's Response to Decision Letter for (RSOS-182122.R0)

See Appendix A.

RSOS-190399.R0

Review form: Reviewer 1

Is the manuscript scientifically sound in its present form?

Yes

Are the interpretations and conclusions justified by the results?

Yes

Is the language acceptable?

Yes

Is it clear how to access all supporting data?

Yes

Do you have any ethical concerns with this paper?

No

Have you any concerns about statistical analyses in this paper?

No

Recommendation?

Accept as is

Comments to the Author(s)

The manuscript is now ready for publication after revision.

Review form: Reviewer 2

Is the manuscript scientifically sound in its present form?

Yes

Are the interpretations and conclusions justified by the results?

Yes

Is the language acceptable?

No

Is it clear how to access all supporting data?

Yes

Do you have any ethical concerns with this paper?

No

Have you any concerns about statistical analyses in this paper?

No

Recommendation?

Accept with minor revision (please list in comments)

Comments to the Author(s)

The authors have addressed many main points but the manuscript still needs to be revised for language.

Decision letter (RSOS-190399.R0)

05-Apr-2019

Dear Dr Wang:

Title: Original Plant Traceability of Dendrobium Species Using Multi-Spectroscopy Fusion and Mathematical Model

Manuscript ID: RSOS-190399

Thank you for submitting the above manuscript to Royal Society Open Science. On behalf of the Editors and the Royal Society of Chemistry, I am pleased to inform you that your manuscript will be accepted for publication in Royal Society Open Science subject to minor revision in accordance with the referee suggestions. Please find the reviewers' comments at the end of this email.

The reviewers and handling editors have recommended publication, but also suggest some minor revisions to your manuscript. Therefore, I invite you to respond to the comments and revise your manuscript.

Because the schedule for publication is very tight, it is a condition of publication that you submit the revised version of your manuscript before 14-Apr-2019. Please note that the revision deadline

will expire at 00.00am on this date. If you do not think you will be able to meet this date please let me know immediately.

Best wishes,
Dr Laura Smith
Publishing Editor, Journals

On behalf of the Subject Editor Professor Anthony Stace and the Associate Editor Professor Hazel Cox.

RSC Associate Editor
Comments to the Author:
(There are no comments.)

Reviewer comments to Author:
Reviewer: 2

Comments to the Author(s)
The authors have addressed many main points but the manuscript still needs to be revised for language.

Reviewer: 1

Comments to the Author(s)
The manuscript is now ready for publication after revision.

Author's Response to Decision Letter for (RSOS-190399.R0)

See Appendix B.

Decision letter (RSOS-190399.R1)

15-Apr-2019

Dear Dr Wang:

Title: Original Plant Traceability of Dendrobium Species Using Multi-Spectroscopy Fusion and Mathematical Model
Manuscript ID: RSOS-190399.R1

It is a pleasure to accept your manuscript in its current form for publication in Royal Society Open Science. The chemistry content of Royal Society Open Science is published in collaboration with the Royal Society of Chemistry.

Appendix A

Dear editor,

Thanks for your kind letter of “**Original Plant Traceability of *Dendrobium* Species Using Multi-Spectroscopy Fusion and Mathematical Model**”. We have revised the manuscript in accordance with the reviewers’ comments, and carefully proof-read the manuscript to minimize errors. The changes we made in the manuscript have shown a new manuscript.

Here below is our description on revision according to the reviewers’ comments.

Response to Reviewer 1

Question: (1) As stated by authors, MIR and UV-Vis spectra was used to authenticate the multi-origin Fengdou crude materials in Ref. [29]. What’s the difference between these two articles? Just two different techniques?

Answer: (1) Thanks for your kind suggestions. There were some differences between two articles in Ref. [29] and our present study. The former focused on the combination between chemical structure information and different chromophores (C=C, C=O) and auxochromes (-OR, -OH), which cover near-infrared spectra range from 10000 to 4000 cm^{-1} and ultraviolet-visible spectra from 600 to 200 nm. The latter was concentrated on complementary NIR and MIR spectra from 10000 to 650 cm^{-1} which reflect more detailed structure information of chemical components within these *Dendrobium* species. In addition, published study of Ref. [29] didn’t contain the most popular species—*D. officinale* which was the main species in the herbal market. What’s more, the fusion strategy with complementary NIR and MIR spectra hasn’t been reported for authentication of the multi-origins Fengdou crude materials.

Question: (2) Line 56, Page 6. The reason for accumulation number was wrong.

Answer: (2) Thanks for your kind suggestions. It’s our wrong explanation to the reason for accumulation scans and the wrong writing has been removed in our revised manuscript. In addition, we have learnt that the accumulation scans were related to signal-to-noise ratio which increase with accumulation time (Stuart B. Infrared spectroscopy, Kirk-Othmer Encyclopedia of Chemical Technology. John Wiley &

Sons, Inc., 2005.).

Question: (3) Why did authors choose to use these three methods?

Answer: Thanks for your kind question for improving our scientific writing. The reason of application of three models is that different datasets with suitable model would generate good discrimination results. Therefore, three methods were investigated find the most suitable classification methods. In order explaining the reason, the advantages and disadvantages of these three methods have been discussed in our revised manuscript.

Random forest depends on the best ntree and optimal split variables (mtry) based on lowest out-of-band (OOB) error. In addition, re-constructed matrix comprised of important variables in light of permutation accuracy importance would influence the final discrimination result.

Support vector machine as a non-linear classification model mainly depends on hyperplane and kernel function. Generally, the model has excellent generalization ability when compared with other multivariate statistical approaches.

Partial least squares discrimination analysis is a variant of partial least squares regression. Different from regression analysis which is used for modeling relationship between dependent variables (Y) and independent variables (X), PLS-DA tries best to find a relationship between variables matrix (X) and label dataset (Y) for prediction of unknown samples. Compared with two methods mentioned above, the method has many evaluation parameters for result discussion.

Finally, each chemometrics method has their respective advantages in terms of theory, calculation time, evaluation method, etc. Therefore, three methods were investigated for comparison of discrimination.

Question: (4) The section of discussion was simple, and the advantages and disadvantages of these three methods should be discussed.

Answer: (2) Thanks for your kind suggestions. We have added a chapter of “Comparison of discrimination performance in three models” in our revised

manuscript.

Comparison of discrimination performance in three models

Based on the results from three models and two fusions strategies above, low-level fusion method was recommended as the optimal fusion method for the discrimination analysis of 12 *Dendrobium* species. It was interpreted that the low-level fusion strategy contains the most original information from different spectra which would reflect the maximum difference among 12 species. As for mid-level fusion method, accuracy rate of calibration set in RF and accuracy rate of validation set in PLS-DA were less than 100%. C value of cross-validation in mid-level fusion was higher than that of low-level in SVM. The poor performance of mid-level fusion could be explained that the variable transform might lose some chemical variable information although it has rapid calculation speed.

Because of the equal discrimination performance of three model with low-level fusion strategy, the advantage of low-level fusion was discussed in the chapter rather than the comparison of three models.

Question: (5) The sample number used for was limited. I wonder the reliability of the results.

Answer: (5) Because 119 specimens from 12 species were obtained from nursery garden of Orchidaceae plants in Wenshan University in Wenshan Prefecture, there aren't enough samples of each species to be collected for the purpose of resource protection. Surely, we will collect enough samples to investigate whether this method is valid on other data in the further study.

Question: (6) How about the results of cross-validation for these three methods?

Answer: It's our negligence that we didn't clearly interpret the conception of cross-validation. In our present study, calibration set was used to replace cross-validation while validation set was equal to the conception of prediction set.

Question: (7) Grid search support vector machine should be replaced by support

vector machine.

Answer: Thanks for your kind suggestions. Grid search support vector machine has been replaced by support vector machine in our revised manuscript. However, grid search was still used because of its advantages [1, 2].

[1] Wang Y, Shen T, Zhang J, Huang H, Wang Y. 2018 Geographical authentication of *Gentiana rigescens* by high-performance liquid chromatography and infrared spectroscopy. *Anal. Lett.* 51, 2173-2191.

[2] Wu Z, Zhao Y, Zhang J, Wang Y. 2017 Quality assessment of *Gentiana rigescens* from different geographical origins using FT-IR spectroscopy combined with HPLC. *Molecules.* 22, 1238.

Question: (8) Line 60, Page 7. The authors stated reflection and transmission mode of mid-infrared spectroscopy were used in this experiments. However, the comparison for these two modes was not presented.

Answer: It's our negligence that we didn't clearly show the comparison for these two modes. In the chapter of "Interpretation of mid-infrared spectra", we have made the comparison between them: Due to the fact that score plots (Figure S2A and S2B) didn't show satisfying cluster performance using transmission FTIR spectra (Figure S3) after pretreatment by SD and MSC, ATR-FTIR spectra (Figure 4) was recommended for further fusion analysis with NIR spectra matrix. Even though the first two PCs interpreted 36.34% original information which was more than that ATR-FTIR spectra (29.25%). In addition, there were similar spectra characteristics just in terms of spectra peaks between 4000 to 1000 cm^{-1} , which has the ability to reflect most of chemical information of these species. In addition, it approximately spent 1 min to scan one sample using reflection mode while transmission mode spent 10 min at least because of the long-time grind with potassium bromide. In addition, it approximately spent 1 min to scan one sample using reflection mode while transmission mode spent 10 min at least because of the long-time grind with potassium bromide.

Question: (9) Language should be further improved. Lots of grammatical mistakes, such as Line 21-23, Page 6; Line 41-44, Page 7; Line 50-51, Page 7; Line 56-57, Page 7.

Answer: Thanks for your kind suggestions and chance to correct these grammatical mistakes.

Line 21-23, Page 6: “Potassium bromide (KBr) was obtained from Tianjin Fengchuan Fine Chemical Research Institute (Tianjin, China)” has been rewritten as “Potassium bromide (KBr) powder was purchased from Tianjin Fengchuan Fine Chemical Research Institute (Tianjin, China)” in our revised manuscript.

Line 41-44, Page 7: “In addition, there are also overlapping absorbance peaks and noisy signal caused interference for further discrimination purpose.” has been rewritten as “In addition, there are also overlapping absorbance peaks and noisy signal which would lead to interference for further discrimination purpose.”

Line 50-51, Page 7: “Therefore, second derivative was used for increasing resolution and amplifying the weak absorbance peak belong to chemical structure” has been rewritten as “Therefore, second derivative was used for increasing resolution and amplifying the weak absorbance peaks”

Line 56-57, Page 7: “Two preprocessing methods and data transform from spectra were computed by SIMCA 14.1 software (Umetrics, Umeå, Sweden).” has been rewritten as “Two preprocessing methods and data transform were computed by SIMCA 14.1 software (Umetrics, Umeå, Sweden).”

In addition, other spell mistakes and grammatical mistakes were also corrected in our revised manuscript such as the redundant spacing.

Response to Reviewer 1

Question: A discussion of the relevant bioactive components and their relation to the finding in the IR data is required. How do the species relate to each other with relevance to the metabolites is also important.

Answer: Thanks for your kind suggestions, we have added the relevant bioactive component and their relation to the finding in the ATR-FTIR spectra in our revised

manuscript. For instance, “Amongst these spectral variables, the absorbance at 3350.70 cm⁻¹ (O-H stretching vibration) combined with band range between 1200 and 1030 cm⁻¹ is responsible for the dominant polysaccharide in these crude materials from *Dendrobium* plants [5,47]. Absorbance peaks between 900 and 650 cm⁻¹ was the fingerprint range of polysaccharide which is main active components within these herbs [7]. Herein, *Dendrobium* polysaccharides was mainly contributed to these bioactivities, such as anti-tumor, anti-diabetic, immuno-modulatory, and antioxidant properties.”

Question: In page 4 you mention that the species are quite different in their metabolites so how can you apply this to your discrimination results?

Answer: Thanks for your kind suggestions. It is the fact that these species are quite different in their metabolites. Compared with common chemistry analysis methods, spectroscopy mainly reflects chemical structure information of most of metabolites, such as symmetrical and asymmetrical stretching vibration belonging to -CH₃ and -CH₂-, O-H stretching vibration, etc. However, single spectroscopy has their respective advantages with parts of metabolites information. Therefore, fusion spectra in our present study including near-infrared and mid-infrared spectra were investigated for discrimination results with more comprehensive chemical information in which not only chemical structure information but their overtones and combinations of fundamental vibrations absorbance hydrogen.

Question: The interpretation in page 12 deals with functional groups rather than prediction of actual metabolites.

Answer: Thanks for your kind suggestions. It is the fact that the interpretation of spectra deals with functional groups instead of prediction of actual metabolites. Because of scientific fingerprint has fuzzy train not only in chromatograms but in spectra (Layloff T. Scientific fingerprinting: a pharmaceutical regulatory tool. Pharmaceutical Technology, 1991.), it is difficult that the spectra peaks interpret actual metabolites. Surely, we still feel great appreciated for your suggestions. In the

further study of *Dendrobium* species, we will try our best to interpret the detailed metabolites in these spectra when we could obtain more chemical standards and have more better analytical technique.

Appendix B

Dear editor,

Thanks for your kind letter of “**Original Plant Traceability of *Dendrobium* Species Using Multi-Spectroscopy Fusion and Mathematical Model**”. We have revised the manuscript in accordance with the reviewers’ comments, and carefully proof-read the manuscript to minimize errors. The changes we made in the manuscript have shown a new manuscript.

Here below is our description on revision according to the reviewers’ comments.

Response to Reviewer 1

Question: The authors have addressed many main points but the manuscript still needs to be revised for language.

Answer: Thanks for your chance to make us revise the manuscript again, our manuscript has been revised in language below:

1. “In a conclusion, low-level fusion strategy **comprised** two spectra after pretreated by second derivative and multiplicative scatter correction was recommended for discrimination analysis because of excellent model performance in three models” in abstract chapter has been corrected as “In a conclusion, low-level fusion strategy **comprised of** two spectra after pretreated by second derivative and multiplicative scatter correction was recommended for discrimination analysis because of excellent model performance in three models”.
2. “Compared with **ATR-FTIR** spectra,” in abstract chapter has been corrected as “Compared with **mid-infrared** spectra,”.
3. “The protocol combined with low-level fusion strategy and chemometrics provides a rapid and effective reference for control of botanical origins **of** crude *Dendrobium* materials” in abstract chapter has been corrected as “The protocol combined with low-level fusion strategy and chemometrics provides a rapid and effective reference for control of botanical origins **in** crude *Dendrobium* materials”.
4. “Especially herbal materials, they have become an important and essentials portion in diet supplements of Asian (China, Korea, Indian and Japan), the United States, Canada, France, **et al** [1]” in introduction chapter has been corrected as

“Especially herbal materials, they have become an important and essential portion in diet supplements of Asian (China, Korea, Indian and Japan), the United States, Canada, France and so on [1]”.

5. “In contrast to these methods, spectroscopy combined with chemometrics is a rapid approach and it has been extensively applied for *Dendrobium* species traceability [17]” in introduction chapter has been corrected as “In contrast to these methods, spectroscopy combined with chemometrics is a rapid approach and it has been extensively applied for traceability of *Dendrobium* species [17]”.
6. “In order to display the daily usage, accurate Latin name and traditional utilization in national minorities of China were summarized in Table 1 [30]” in Materials and methods chapter has been corrected as “In order to display the daily usage, accurate Latin name and traditional utilization in national minorities of China were summarized in Table 1”.
7. “An Antaris II spectrometer (Thermo Fisher Scientific, USA) with integrating sphere diffused reflection mode was used for obtaining near-infrared spectra (NIR) of sample powder” in Materials and methods chapter has been corrected as “An Antaris II spectrometer (Thermo Fisher Scientific, USA) with integrating sphere diffused reflection mode was used for obtaining NIR spectra of sample powder”.
8. “Next measurement was conducted after drying avoiding the interference between different species or different individuals from the same species” in Materials and methods chapter has been corrected as “Next measurement was conducted after drying to avoid the interference between different species or different individuals from the same species”.
9. “Briefly, the refined powder (1.2 mg) of each sample was sufficiently ground with KBr for thin and hyaline tablet” in Materials and methods chapter has been corrected as “Briefly, the refined powder (1.2 mg) of each sample was sufficiently ground with KBr for obtaining thin and hyaline tablet”.
10. “two kinds of spectra after pretreatment were directly connected with each other, the number of rows were the same as the number of samples while the number of columns were equal to the total of variables information from two kinds of

spectra.” in Materials and methods chapter has been corrected as “two kinds of spectra after pretreatment were directly connected with each other. The number of rows were the same as the number of samples while the number of columns were equal to the total of variables information from two kinds of spectra”.

11. “Generally, the model has excellent generalization ability when compared with other multivariate statistical approaches [43]” in Materials and methods chapter has been corrected as “Generally, the model has excellent generalization ability compared with other multivariate statistical approaches [43]”.
12. “The best combination of the two parameters was based on the highest accuracy rate” in Materials and methods chapter has been corrected as “The best combination of the two parameters is based on the highest accuracy rate”.
13. “Besides of accuracy rates of calibration and validation set, the evaluation of model performance is also provided by sensitivity, specificity, precious and efficiency” in Materials and methods chapter has been corrected as “Besides of accuracy rates of calibration and validation set, the evaluation of model performance was also provided by sensitivity, specificity, precious and efficiency”.
14. $\text{Efficiency} = \sqrt{\text{sensitivity} \times \text{specificity}}$ was rewritten as $\text{Efficiency} = \sqrt{\text{Sensitivity} \times \text{Specificity}}$.
15. “As the results of parameters optimization indicated in Figure 11, excellent discrimination results (Table 6) were obtained with $c = 2$ and $g = 0.0652$ that both accuracy rates of calibration and validation sets were 100%” was rewritten as “As the results of parameters optimization indicated in Figure 11, excellent discrimination results (Table 6) were obtained with $c = 2.000$ and $g = 0.0652$ that both accuracy rates of calibration and validation sets were 100%”.
16. In addition, some errors in the reference list were also revised.
17. “Figure 11 Results of GS-SVM method with mid-level fusion strategy” was rewritten as “Figure 11 Results of SVM-GS method with mid-level fusion strategy”.

Response to Reviewer 2

Question: The manuscript is now ready for publication after revision.

Answer: Thanks for your chance to make us revise the manuscript again, our manuscript has been revised in which the detailed points were displayed above “Response to Reviewer 1”.